# Three-Dimensional Semantic Segmentation of Palatal Rugae and Maxillary Teeth and Motion Evaluation of Orthodontically Treated Teeth Using Convolutional Neural Networks

**DOI:** 10.3390/diagnostics15111415

**Published:** 2025-06-02

**Authors:** Abdul Rehman El Bsat, Elie Shammas, Daniel Asmar, Kinan G. Zeno, Anthony T. Macari, Joseph G. Ghafari

**Affiliations:** 1Department of Mechanical Engineering, Maroun Semaan Faculty of Engineering and Architecture, American University of Beirut, Beirut P.O. Box 11-0236, Lebanon; ahe23@mail.aub.edu (A.R.E.B.); es34@aub.edu.lb (E.S.); da20@aub.edu.lb (D.A.); 2Department of Dentofacial Medicine, Faculty of Medicine, American University of Beirut, Beirut P.O. Box 11-0236, Lebanon; kz12@aub.edu.lb (K.G.Z.); jg03@aub.edu.lb (J.G.G.)

**Keywords:** three-dimensional imaging, orthodontics, convolutional neural networks, palate, machine learning, artificial intelligence

## Abstract

**Background:** The segmentation of individual teeth in three-dimensional (3D) dental models is a key step in orthodontic computer-aided design systems. Traditional methods lack robustness when handling challenging cases such as missing or misaligned teeth. **Objectives:** to semantically segment maxillary teeth and palatal rugae in 3D textured scans using Convolutional Neural Networks (CNNs) and assess tooth movement after orthodontic treatment using stable rugae references. **Methods:** Building on the robustness of two-dimensional image semantic segmentation, we developed a method to convert 3D textured palate scans into two-dimensional images for segmentation, then back projected them onto the original 3D meshes. A dataset of 100 textured scans from 100 patients seeking orthodontic treatment was manually segmented by orthodontic experts. The proposed 3D segmentation method was applied to these scans. Finally, each pair of segmented 3D scans from the same patient, before and after treatment, was aligned by superimposing them on the stable rugae region. **Results:** The 3D segmentation method achieved an accuracy of 98.69% and an average Intersection over Union (IoU) of 84.5%. The common stable coordinate frame for both scans using the rugae area as a stable reference enabled the computation of the 3D translational and rotational motions of each maxillary tooth. Neither pre- nor post-processing of the data was required to enhance segmentation. **Conclusions:** The proposed method enabled successful motion measurement of teeth using the rugal area as a stable reference and providing rotation and translational measurements of the maxillary teeth.

## 1. Introduction

The main goal of digital dentistry is to replace cast impressions with 3D digital models for treatment planning and estimation of tooth movement. To this end, the 3D meshes must be segmented to identify the individual teeth and superimpose sequential scans on relatively stable oral anatomic structures. This process enables the accurate measurement of tooth movement during growth and orthodontic treatment. Considering the variability in tooth shapes, the development of a general robust geometric model is essential to define the parameters of teeth and segment them individually through machine learning methods. Several authors have attempted to reach this goal, but their methods had various limitations, including human intervention and/or pre-processing or post-processing of the data [1,2,3,4,5].

Recent developments in deep learning have advanced the segmentation of 3D dental meshes. Krenmayr et al. [1] proposed DilatedToothSegNet, enhancing segmentation accuracy through expanded receptive fields. Wang et al. [2] introduced STSN-Net, which effectively segments and numbers teeth, even in crowded conditions. Estai et al. [3] demonstrated successful automated detection and numbering of permanent teeth on panoramic images, supporting the growing role of AI in dental diagnostics.

Despite this progress, challenges remain with partial or low-quality scans. Jana et al. [4] highlighted the limitations of current deep learning models in handling incomplete dental meshes. Raith et al. [5] showed that artificial neural networks can classify 3D tooth features effectively, but remain sensitive to data quality. These findings suggest that, while AI tools hold significant promise, further refinement is needed for broader clinical use.

This study aimed to develop a fully automated tooth segmentation algorithm based on machine learning for evaluating dental movement following orthodontic treatment. To achieve this, a labeled dataset of 3D texture-colored models from real patients, along with their corresponding 2D projections, was created.

## 2. Methods

The planned strategy consisted of segmenting 3D textured scans, first by performing 2D semantic segmentation of 2D images using a robust Fully Convolutional Neural Network (F-CNN), then back projecting onto the original 3D scans. The motions of individual orthodontically displaced teeth would then be measured (via the Iterative Closest Point (ICP) method) after defining a stable coordinate frame on geometrically segmented palatal rugae, used as stable references to match the scans. The motion estimate would then be decomposed into unique three-dimensional translation and rotation about the unique axes of the coordinate frame.

### 2.1. Deep Network Architecture

A deep network architecture was devised to perform semantic segmentation of 2D images using an Encoder–Decoder–Skip based on SegNet [6,7], which relies on a VGG-style encoder–decoder. Because the regions of teeth and rugae appear as continuous blobs in the 2D images, an attention layer was designed and introduced into the network to improve training accuracy.

The encoder network consists of 13 convolutional layers. For each layer, a convolution with a filter bank produced a set of feature maps. This step was followed by batch normalization and an element-wise ReLU; then, max-pooling was performed with a 2 *×* 2 window and stride of 2. For every encoder layer, there was a corresponding decoder layer, for a total of 13 layers, similar in structure to the encoder layers (Figure 1). While the latter use max pooling, the decoder layers use up-sampling of the input feature map, followed by batch normalization. The up-sampling in the decoder uses transposed convolutions. The architecture employs additive skip connections from the encoder to the decoder layers. To improve the accuracy of semantic segmentation, two attention layers, (channel and spatial) were implemented, and their effects were evaluated. The channel attention aims to learn a one-dimensional weight and assign it to a corresponding channel by utilizing the relationship between each channel of the feature map. Spatial attention uses the relationships between different spatial positions to learn and assign a 2D spatial weight to a corresponding position. Adding the attention layers helps the model to learn more representative features.

Two different implementation augmentations were tested on the architecture. In the first, the feature map was fed into the channel and spatial attention layers before being fed into the final (Softmax) layer that generates the predictions [8,9,10]. In the second application, the channel and spatial attention layers were applied at the transition between the encoding and decoding layers. The training prediction results are presented in Section 2.2, below.

### 2.2. 3D Data Collection and Annotation

The material comprised 100 3D textured scans of the maxilla that were manually processed to remove disconnected components, and the back base was constructed. In each 3D mesh, the surface areas that represent the individual teeth, the rugae, and the gingiva were manually labeled. The meshes were then rotated, translated, and positioned in roughly the same configuration in 3D space. This step was followed by the 2D projection of the meshes onto a virtual camera image plane; 100 random 3D camera poses were used for each scan. The lighting conditions were also randomized to generate the 2D images.

#### 2.2.1. Dataset Collection

One hundred maxillary arch scans were randomly selected from patient records, regardless of age or gender. A dataset was generated, labeled, and augmented from the set of texture-colored 3D meshes of the maxilla (Figure 2A). The 3D images were taken, using a 3D intraoral scanner (3Shape TRIOS Intraoral scanners—Copenhagen, Denmark) in the Department of Dentofacial Medicine at the American University of Beirut Medical Center, and saved as textured 3D meshes using the PLY file format.

#### 2.2.2. 3D Mesh Shell Decomposition

The 3D meshes used for semantic segmentation were annotated by assigning a class to every vertex in the mesh and were manually segmented using Meshmixer [11]. In this process, the user can select the vertices that represent an individual element and separate the selected shell into an independent component. Manual selection often results in rough edges because of limitations of human precision. Therefore, a built-in boundary smoothing tool was applied before finalizing the selected regions for separation, ensuring a more accurate and visually coherent segmentation. The procedure was performed on a total of 100 3D scans, generating independent components of the teeth and rugae patch (Figure 2B–D).

#### 2.2.3. 3D Mesh Shell Labeling

The vertices in each component were assigned to a class by color using a fill coloring tool in MeshLab version 2016.12 [12,13], such as coloring the right central incisor with RGB [85,85,0] (Figure 2E). Accordingly, 28 unique labels were generated: 26 for teeth, 1 for rugae, and void. The gingiva and background were assigned a black color, denoting a void label.

#### 2.2.4. 3D Mesh Label Statistics

To assess the variability of label size, the statistics of the label distribution throughout the dataset were computed. For every 3D scan, 100 projection images were produced. To ensure that the projections were not biased, the relative number of vertices for each label should relate to the corresponding number of pixels. Most of the low labels fell in the primary tooth and third molar categories, as the dataset had minimal representation of these labels.

The distribution of the number of pixels per label is the same as the distribution of the number of vertices per label. To verify this consistency, the number of pixels per individual label was divided by the number of vertices for that label, and the ratio was computed for all labels. The average ratio was 25 ± 4.4 SD, and the variation in the ratios was minimal because of the randomness of the projection viewpoints, as well as the size of the label.

### 2.3. Generating Labeled 2D Data

Using simulated camera projection, 2D images were generated from the labeled 3D meshes. For each 3D mesh, an algorithm was used to generate 100 labeled 2D images, which were used to train a 2D semantic segmentation convolution network.

#### 2.3.1. Coordinate Frame Definition

When 3D scans are initially generated, their orientation is not calibrated to a fixed reference in space. Hence, a standardized coordinate frame was established to enable a robust and consistent alignment across all scans. This process ensures that all scans are spatially normalized, allowing for accurate relative comparisons.

The coordinate frame was defined by manually placing 6 spheres on specific anatomical landmarks of the mesh (Figure 3A). The spheres were positioned at the molar cusps, at the incisor edges, and along the median raphae in the rugae region. Two planes could be generated: the occlusal plane, determined by the centers of the three spheres located at the cusps of the molars and incisors, and the sagittal plane, defined by the alignment of the spheres located along the median raphae, through the middle of the rugae patch [14]. A coordinate frame was defined with the *z*-axis normal to the occlusal plane, and the *x*-axis along the line of intersection of the occlusal and sagittal planes. The origin of the coordinate frame was defined as the normal projection of the center of the sphere located at the incisive papilla onto the occlusal plane. The *y*-axis was drawn to complete a right-handed orthonormal frame, ensuring consistency across all scans (Figure 3B).

#### 2.3.2. 2D Dataset Generation

A simulated camera projection was used to transform the 3D textured meshes into 2D images, taken from various viewpoints. The 100 viewpoints were distributed randomly on a hemisphere with the camera’s optical axis pointing toward the 3D scan. The vertical axis of the camera was randomly rotated about its optical axis to randomize the rotation of the mesh within the 2D projections. The projections were applied to the textured 3D mesh and the associated 3D labeled mesh. Accordingly, for each 3D mesh, this method produced 100 pairs of 2D images, along with their associated labels. The flowchart depicting the process is shown in Figure 4. This labeling method not only eliminated the time needed to hand label the 2D images, but also produced accurate and consistent labels for all classes.

#### 2.3.3. Data Augmentation

The 2D image generation was augmented to account for scenes under various lighting conditions: glow with no lighting, natural ambient light at infinity, and close spot lights with varying intensity and direction. For the last of these, two positions were chosen for light location, but the light intensity was randomly generated at every iteration. The three lighting conditions varied in the way the software rendered the shadows on the 3D mesh, with the first eliminating the shadows completely, the second producing consistent parallel shadows, and the third generating variable shadows, depending on the spot light position with respect to the mesh.

Lighting was not used for the associated 3D labeled mesh; instead, the colors were assigned as glowing colors. Moreover, the anti-aliasing option of the shader was turned off, ascertaining the lack of color mixing at the boundary between two labels. Another augmentation was implemented that randomized the perspective projection of the camera model, thereby simulating the mesh projection at various perspective distortions. For each 3D mesh in the original 3D dataset, 100 2D images were created with their associated labels. Thus, the 2D dataset of images included 10,000 labeled images. A sample of the images, various data augmentations, and the associated image labels are shown in Appendix A (Figure A1). 

### 2.4. Rasterization

The original 3D meshes were segmented through 2D semantic segmentation using back projection. Starting with a 3D textured mesh, a set of 2D images was generated by simulating a set of known camera positions to capture the entire 3D mesh, then segmented using the trained model. Subsequently, using the known camera positions, the image predictions were used to back project the label onto the 3D mesh. The predictions on the images, known as rasters, could be back projected (or baked) onto the mesh using the saved camera poses. Afterwards, all vertices were assigned a label according to their baked colors, before segmenting the mesh by color to achieve the 3D semantic segmentation of the entire mesh (Figure 5).

### 2.5. Motion Estimation

After performing 3D segmentation on two meshes from the same patient, an algorithm was developed to measure the motion of all maxillary teeth. The first and second scans were the pre-treatment and post-treatment scans, respectively. The alignment of the scans on the defined coordinate frames provided rough initial positioning, insufficient for accurately estimating the motion of individual teeth. Aligning the pre- and post-treatment scans was further complicated because of the composition of the mesh, which includes the soft tissue (mucosa), which undergoes deformation, and the rigid teeth that had moved between scans. As a result, direct global alignment would lead to inaccurate captures of tooth motions. The rugae were reported as geometrically stable structures. Specifically, changes at the medial rugae points were less clinically significant than those at the lateral points [15,16]; thus, they were ideal references for alignment. Hence, the semantic segmentation included the rugae patch as a stable landmark upon which the pre- and post-treatment scans were aligned to measure the relative tooth movement.

The rugae patch of the post-treatment scan was first aligned to the pre-treatment scan using the Iterative Closest Point (ICP) algorithm. The metrology software Polyworks version 2017 (InnovMetric Software Inc., Quebec City, QC, Canada) [17,18] was used to perform the ICP algorithm and mesh transformations). The algorithm generates a transformation matrix that represents a 3D rigid body motion. Applying this transformation to the post-treatment scan transforms the entire scan and aligns it relative to the rugae patch (Appendix A Figure A2A,B). Estimating the motion of the individual teeth entailed separately aligning every single tooth with its counterpart from the other mesh. After the first ICP implementation alignment of the scans, a second implementation was applied to each tooth individually (Appendix A Figure A2C). The rigid body transformation output of the second implementation isolated the individual tooth movement. Accordingly, for each tooth, an associated transformation matrix that gauged the tooth motion was computed.

## 3. Method Evaluation and Results

The machine learning algorithms applied to the 2D projection and the final results of the 3D segmentation of the original meshes are presented in this section.

### 3.1. Training Evaluation

Semantic segmentation predictions are generally evaluated using average mean Intersection over Union (average mIoU), which is the mean of all the IoU values for all the classes in an image pair, ground truth and prediction. For a dataset comprising many image pairs, the average mIoU is the average of the mean IoUs for all image pairs. The IoU for a given class *c* can be defined [19,20] as follows:(1)$$IoU\left(c\right)=\frac{\sum_{i}o_{i}==c\;⋂y_{i}==c}{\sum_{i}o_{i}==c\;⋃y_{i}==c}$$
whereby *o_i_* is for prediction pixels, *y_i_* for target or label pixels, ∩ is a logical and operation, and ∪ is a logical or operation. This equation is summed over all of the pixels *i* in the image.

Moreover, since the dataset exhibits class imbalance, that is, the various class sizes are not similar, the average mIoU represents a better metric through which to assess the network compared to the pixel accuracy because mIoU gives the same weight to under-represented classes as to all other classes. The class imbalance in the dataset exists because the background and the rugae labels cover relatively larger areas compared to the rest of the classes, namely the teeth. Hence, the high pixel accuracy does not imply superior segmentation ability [21].

### 3.2. Attention Layer Evaluation

To gauge the effects of the two attention layers (channel and spatial), a *reduced dataset* of 2000 images (20 scans with 100 projections each) was used to lower the computation time and determine the attention layer implementation to use on the full dataset. The latter was split into 76.4% for training (1528 images), 7.0% for validation (139 images), and 16.6% for testing (333 images). The hyper parameters used for the architecture training were as follows: 150 epochs, a batch size of 1, a learning rate of 0.0001, and a decay of 0.995.

To assess the value of all three architectures (original and both attention implementations), the architectures were trained on the reduced dataset. The training results are shown in Table 1, including prediction results conducted on unseen projections of unseen 3D scans. The best results were obtained when placing the attention layers at the end, with an average accuracy of 98% and an average mIoU of 79.01%, which was expected because the model was trained on 20 scans, a relatively small number for training (Figure 6). Therefore, this implementation was used to refine label prediction [8,9,10].

### 3.3. Final Network Evaluation

The dataset of 100 scans was split into 80% for training (80 scans with 8000 associated projection images), 10% for validation (10 scans, 1000 associated projection images) and 10% for testing (10 scans, 1000 associated projection images). The validation and test scans, and their associated projections, were disjointed from the 80 scans used for training. The 80/20 training–test split is a common practice in machine learning. This choice ensures that the model is trained on a larger portion of the data, allowing it to better capture feature variability and learn complex patterns. The increased training size also helps to reduce overfitting and improves generalization on the test set. The hyper parameters used for the architecture training were as follows: a batch size of 1, a learning rate of 0.0001, and a decay of 0.995.

In a sample of the prediction results (Figure 7), the worst, average, and best prediction outcomes are shown in the rows A, B, and C, respectively. The training results are shown in Table 2, in which a precision metric is defined as the ratio of all of the correctly detected pixels with respect to the predicted pixels. Certain labels have low accuracy compared to others because they were not as common in the dataset, such as the labels of primary teeth and third molars. The final training had an average accuracy of 98.69%, an average precision of 98.71%, and an average mIoU score of 85.41%. To assess the network accuracy on the labels found in most scans, the adult teeth average mean IoU was introduced, similar in calculation to the average mean IoU score, but excluding third molars, primary teeth, rugae, and background. The average mean IoU score was 84.26%.

#### 3D Mesh Segmentation Evaluation

The rasterization and final 3D mesh color are depicted. After rasterizing all 10 3D meshes from the test set, the IoU metric was applied to the vertices of the 3D mesh. The results of the 3D mesh segmentation are shown in Appendix A (Table A1), where the mean IoU per scan is listed in the last row. The values are comparable to the 3D average mIoU; however, two meshes (6 and 8) exhibited lower segmentation scores.

In Mesh 6, the lateral incisors were falsely predicted as permanent rather than primary incisors, and the left lateral incisor was still erupting, which made it difficult to predict (Appendix A Figure A3). The low number of erupting teeth in the dataset explains their low prediction accuracy. A similar outcome was observed for the erupting left first pre-molar in Mesh 8, in which the right canine was also falsely predicted as a primary canine because of the close resemblance between primary and permanent canines. Mesh 7 was the best mesh, segmented with an average IoU of 87.67%; the segmentation was very close to the ground truth in all classes. Disregarding rugae, third molars, and Meshes 6 and 8, the proposed method achieved an average IoU value of 84.5%. The rugae exhibited lower segmentation scores compared to teeth because of the variability of the ground truth labeling; labels were manually segmented and classified by several technicians. The projections for the 3D segmentation evaluations were conducted using MeshLab [12,13], whereas the projections used for training the network were performed with Mathematica version 11.0.1 [22]. Given that each software used different shaders to generate 2D images from textures 3D meshes, the prediction results of the MeshLab projections were expected to be of lower accuracy.

### 3.4. Teeth Motion Evaluation

The motion of each individual tooth between pre- and post-treatment scans for the same patient was calculated. To validate the proposed method to compute the rigid motion of 3D meshes, an example was set up with known motion of the segmented parts. Then, the proposed ICP algorithm was applied to gauge whether the known motions were computed accurately. The result of the ICP algorithm is a 4 *×* 4 homogeneous transformation matrix, *H*, which represents a general 3D rigid body motion, and is composed of six individual transformations, three translations, *T_i_*, and three rotations, *R_i_*. The sequence of the six transformations is provided as follows:


(2)
$$H=T_{z}T_{y}T_{x}R_{z}(\alpha )R_{y}(\beta )R_{x}(\gamma )=\left(\begin{array}{cccc} R_{11} & R_{12} & R_{13} & T_{x}\\ R_{21} & R_{22} & R_{23} & T_{y}\\ R_{31} & R_{32} & R_{33} & T_{z}\\ 0 & 0 & 0 & 1 \end{array}\right)$$


The ICP algorithm provides numerical values for *H*, specifically for variables *R_ij_*, *T_x_*, *T_y_*, and *T_z_*. To determine the corresponding transformation parameters, an analytical expression of the transformations is equated to the numerical values of *H*. Using inverse kinematics, the parameters are found as follows:


(3)
$$\beta ={-\sin}^{-1}(R_{31})$$



(4)
$$\gamma ={-\tan}^{-1}(\frac{R_{32}}{R_{33}})$$



(5)
$$\alpha ={-\tan}^{-1}(\frac{R_{21}}{R_{11}})$$


Note that the rotation around *y*-axis *β* has two solutions; however, considering the clinical context of orthodontic applications, a rotation exceeding 90 degrees is implausible for tooth movement. Therefore, the second larger angle solution is excluded to ensure relevant motion estimation.

For the verification example, a segmented pre-scan was used, whereby all of the teeth were transformed manually via a predefined rigid body transformation. These known transformations were labeled “Actual”, and the computed motions using the proposed algorithm were labeled “Experimental” (Table A2 and Table A3). The algorithm was able to identically reproduce the ground truth transformation for all 14 teeth, except for 3 minimal numerical translation errors. The ground truth rotation was computed accurately using the proposed algorithm.

To compute the movement of individual teeth after superimposition on the palatal rugae patch, each tooth pair in the two scans was aligned using transformation alignment, and the resulting transformation was estimated. After verifying that the algorithm accurately computed the rigid body motions of 3D meshes, it was applied to an actual scan pair. Since the real movement of the teeth is unknown, ground truth for the rigid body transformations was not present. The pair of pre-treatment and post-treatment scans for the same patient are shown in Appendix A (Figure A4). Computed rigid body motion for all of the teeth is also depicted in Appendix A (Table A4).

## 4. Discussion

The proposed automated method algorithm was successful in segmenting permanent teeth in 3D, with lesser accuracy on unerupted, primary, and wisdom teeth, and enabled the measurement of rotational and translational tooth movement after orthodontic treatment relative to the palatal rugae used as a stable reference.

This method offers advantages relative to prior studies. Zhao et al. [23,24,25,26] used minimum curvature in 3D scans of the maxillary jaw to initiate the segmentation process, but their method required user interaction at multiple stages to exclude the undesirable areas picked by the curvature-based algorithm. Other authors used artificial neural networks to classify teeth features from a 3D scan, manually labeling only cusps of the teeth, not segmenting the individual teeth [5,27].

Xu et al. [28,29,30,31] performed segmentation directly on a 3D model through classification of mesh faces on a two-level segmentation, the first of which separated the teeth from the gingiva, while the second segmented individual teeth. A label optimization was introduced after each prediction to correct wrongly predicted labels, but sticky teeth (adjacent pairs of teeth with the same label after optimization) were sometimes predicted falsely. In our method, the need for pre-processing and post-processing steps was removed. In addition, Xu et al. did not use a stable reference for motion estimation.

Cui et al. [32,33] performed 3D segmentation using 3D dental scans and cone beam computed tomography (CBCT) images, whereby edges were extracted using a deep network model as a pre-processing step. This method required pre-processing steps to facilitate the segmentation process of the 3D model and another imaging modality, the CBCT images. In addition to removing pre-processing steps, our method enabled the acquisition of textured meshes more frequently in multiple stages without the need for CBCT scans that must be limited due to exposure measures. Tian et al. [34,35,36] involved performing 3D segmentation using three-level hierarchical deep learning. The method also required pre-processing and post-processing of the dental model to enhance the segmentation results achieved, as well as point cloud reconstruction, which could generate non-original mesh data.

One of the most practical methods to assess tooth movement during growth and orthodontic treatment has involved the use of the palatal rugae as a relatively stable reference for superimposition, allowing for multiple stages of assessment and foregoing the need for radiologic exposure. Such approaches have included the generation of a reference coordinate system through the manual transfer of palatal “plugs” across sequential maxillary arches, the labeling of points of reference on the rugae across the arches over time, and the best fit of palatal surfaces that did not necessarily accurately reproduce the stable rugae [15]. These approaches carry potential errors in measurement of tooth movement, particularly rotation and translation. The proposed scheme provides more accurate assessment because of the closer representation of rugal anatomy. However, additional work is needed to use specific pairs of rugae that are more stable over time, as rugal anatomy may be altered by certain tooth movements [15]. Recent evidence suggests that the third rugae are the most stable [37].

### 4.1. Medical Diagnostic Relevance

Accurate segmentation of maxillary teeth and palatal rugae in 3D scans has significant implications for medical diagnostics, particularly in orthodontics, forensic odontology, and craniofacial anomaly assessments. The precise identification of tooth position, movement, and morphology allows for the improved diagnosis of malocclusions, temporomandibular joint disorders (TMDs), and dental asymmetries [38]. Furthermore, this method enhances orthodontic treatment planning by providing quantitative measurements of tooth displacement over time, allowing for personalized interventions and monitoring of treatment efficacy [39]. In forensic sciences, the detailed segmentation of palatal rugae, considered unique to individuals, can serve as a reliable biometric marker for human identification [40]. Additionally, the ability to assess morphological changes in dentition and surrounding structures can aid in the early detection of pathological conditions such as periodontitis and alveolar bone loss [41]. By integrating deep learning-based segmentation with diagnostic applications, this approach holds promise for enhancing clinical decision making and reducing reliance on invasive imaging techniques, thereby improving patient outcomes in dental and orthodontic treatments [42,43].

### 4.2. Limitations and Future Considerations

The proposed method showed excellent performance in 3D semantic segmentation and motion analysis, with high accuracy and no need for additional processing steps. However, there are several opportunities to further strengthen and expand its clinical applicability. While our dataset of 100 scans provided a solid foundation for model development and testing, Future studies with larger and more diverse patient populations should aim to include a broader range of ages, malocclusion types, and dentition stages to better assess and confirm the generalizability of the method. The current approach also offers the potential for comparison with other segmentation techniques in the literature, which could further highlight its efficiency across different clinical scenarios. Additionally, applying the method to the early stages of treatment could reveal its value in detecting changes and improving treatment planning. This method holds strong potential as a tool for evaluating orthodontic treatment quality, and we encourage further validation in broader clinical contexts to fully realize its benefits.

## 5. Conclusions

A maxillary teeth dataset of textured 3D mesh consisted of 100 texture-colored and segmented scans that were used to generate a total of 10,000 labeled projections. A machine learning method with attention layers was used to semantically segment 2D projection images. The best network yielded an accuracy of 98.69% and average mIoU of 85.41. Using rasterization and texture baking, the 2D predictions were used to segment individual teeth and rugae in three-dimensional textured scans.

The proposed method did not require pre- or post- processing of the data to enhance the segmentation. Motion measurements were successfully performed using the palatal rugae area as stable reference points. The method provided rotation and translation measurements of teeth.

## Figures and Tables

**Figure 1 diagnostics-15-01415-f001:**
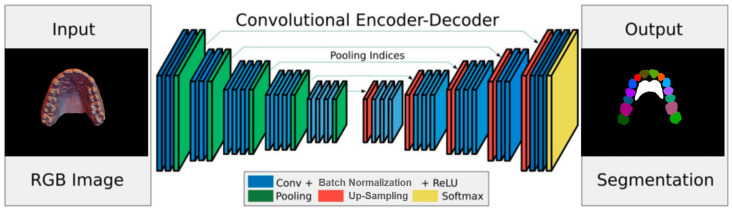
Overview of SegNet architecture. The decoder up-samples using encoder pooling indices, followed by convolution to densify features, with final pixel-wise classification via a softmax layer.

**Figure 2 diagnostics-15-01415-f002:**
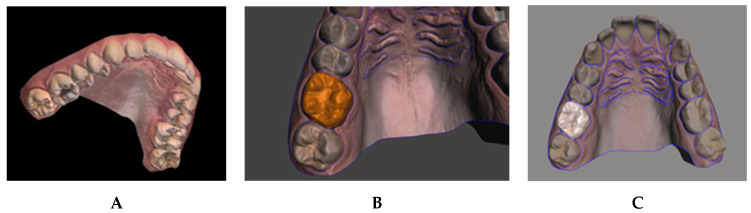
(**A**) 3D scan samples; (**B**) selection of vertices; (**C**) separation of selected objects; (**D**) segmented mesh; and (**E**) labeled mesh.

**Figure 3 diagnostics-15-01415-f003:**
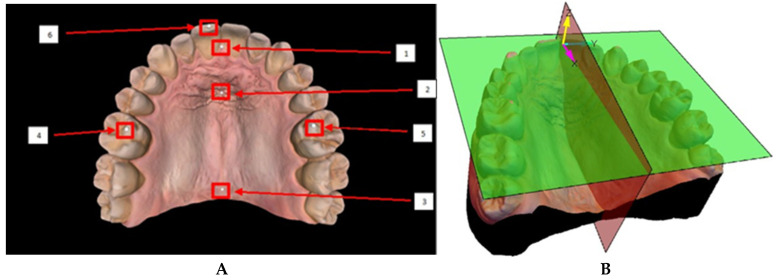
(**A**) Mesh with sphere placement and (**B**) generated coordinate frame.

**Figure 4 diagnostics-15-01415-f004:**
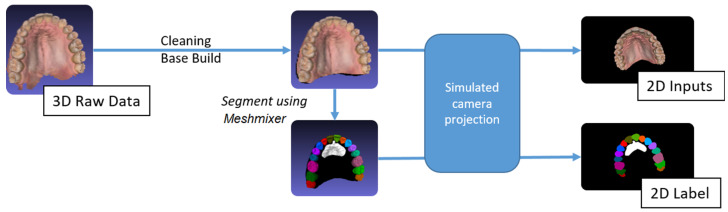
Flow chart of the annotation and 2D projections.

**Figure 5 diagnostics-15-01415-f005:**
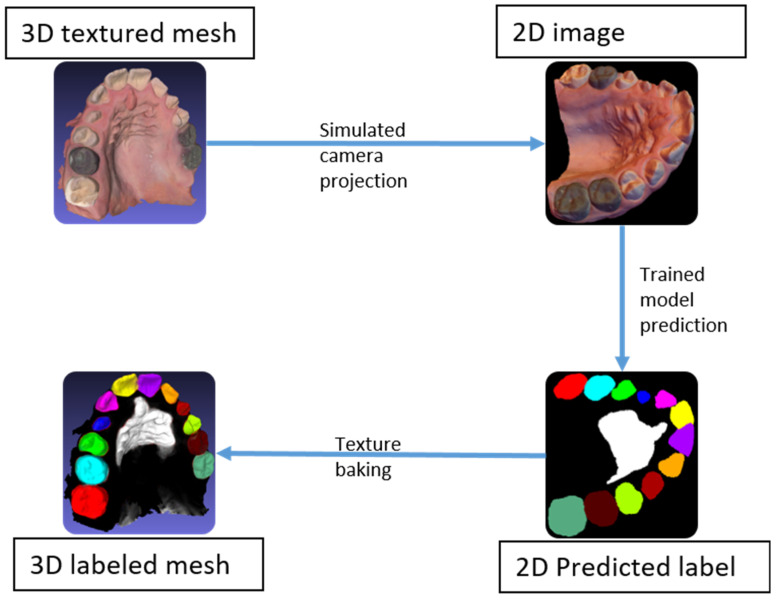
Flow chart of the 3D segmentation process of an unseen textured mesh.

**Figure 6 diagnostics-15-01415-f006:**
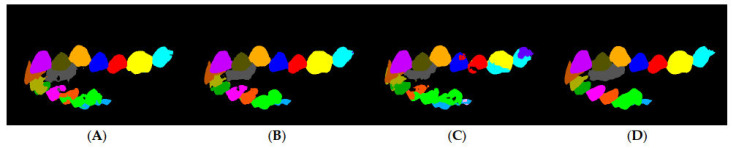
Attention layer comparison on unseen projection of an unseen scan. (**A**) Original SegNet; (**B**) SegNet—Attention (end); (**C**) SegNet—Attention (center); and (**D**) ground truth.

**Figure 7 diagnostics-15-01415-f007:**
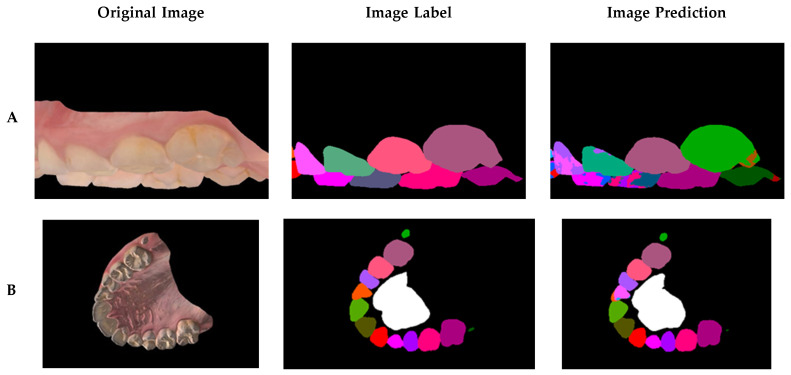
Final training results displayed in rows: (**A**) worst result; (**B**) average result; and (**C**) best result. The average result demonstrates a fairly high accuracy and is close in performance to the best case, indicating consistent behavior. The worst result, while visibly poorer, can be interpreted as an outlier; its contrast against the average and best results highlights the model’s overall robustness and reliability.

**Table 1 diagnostics-15-01415-t001:** Model architecture comparison on unseen scan set.

Model	SegNet Original	SegNet with Attention Layers (End)	SegNet with Attention Layers (Centered)
Avg. Accuracy	97.80	98.00	97.60
Avg. mean IoU score	76.68	79.01	76.35

**Table 2 diagnostics-15-01415-t002:** Model training results on the final dataset.

Model	Mean IoU Scores
left central incisor	94.12
left lateral incisor	78.93
left canine	87.23
left first bicuspid	85.71
left second bicuspid	84.66
left first molar	86.26
left second molar	72.72
left third molar	66.16
right central incisor	94.76
right lateral incisor	83.72
right canine	81.47
right first bicuspid	89.19
right second bicuspid	89.52
right first molar	83.12
right second molar	68.22
right third molar	70.91
rugae	85.87
left primary central incisor	-
left primary lateral incisor	0.00
left primary canine	80.10
left first primary molar	59.21
left second primary molar	77.51
right primary central incisor	-
right primary lateral incisor	0.05
right primary canine	73.27
right first primary molar	72.57
right second primary molar	77.25
background	99.34
Average Accuracy	98.69
Average Precision	98.71
Average mean IoU score	85.41
Adult Teeth Average Mean IoU Score	84.26

## Data Availability

Data are available upon request to interested researchers.

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
