# Peer review of "Three-Dimensional Semantic Segmentation of Palatal Rugae and Maxillary Teeth and Motion Evaluation of Orthodontically Treated Teeth Using Convolutional Neural Networks"

_diagnostics, 2025, doi:10.3390/diagnostics15111415_

Round 1
Reviewer 1 Report
Comments and Suggestions for Authors
Purpose and Methods:
The purpose of the study and the methods used are clearly and consistently defined. The 3D semantic segmentation approach performed with Convolutional Neural Networks (CNN) can be used for orthodontic treatment monitoring. The method converts 3D textured scans into 2D images and labels them by segmentation, then the 3D segmentation structure is obtained. It can be evaluated as a new, original study. The 98.69% accuracy rate and 84.5% IoU value obtained show the success of the proposed segmentation method.
However;
- The explanations of the methods are sufficient, but adding real-world data and more test samples would be useful, and testing the accuracy of the model on a larger data set would be useful.
- The diversity of the data set and comparison of the segmentation techniques used with other literature could also be added. For example, additional explanations could be made on how efficient the same approach is with different tooth types (e.g. primary teeth, early tooth cycles).
- Considering that the obtained result was obtained with a specific dataset and model, it is recommended to test it on different patient groups and different treatment processes.
- In addition, more studies on evaluating tooth movements in early stages may enable this method to provide more clinical benefits.
General Evaluation:
The study takes an important step in measuring and segmenting tooth movements. However, there are some limitations such as small errors and low class accuracies. For example, more data may be needed for segmentation and accurate movement tracking of elements such as early shed teeth or developing teeth. Improvements made in this direction may enable the algorithm to produce more reliable results in the clinical field.
Reviewer 2 Report
Comments and Suggestions for Authors
Dear Authors,
Thank You for a pleasure to read Your article.
I have several comments to improve Your manuscript.
Title
Please, write the type of study.
Abstract
Please, check the rules of journal for words number.
Please, write number of patients (1 or 100).
Please, check keywords according to MeSH.
Introduction
This section is too short, please, enlarge it, for example, with detailed description of previous study (ref.1-5)
The aim of one study could be only one but there are several tasks. Please, choose what is the aim and what are the tasks
Lines 52-58 are more appropriate for materials and methods
Materials and methods
Legend 1. Please, add explanation. Also, please, check the quality of image as it is not clear especially after light zoom.
Dataset collection
Please, add description because it is not clear the number of men, women, children, there is no age or their diagnosis.
As I know the dataset must be maximally clear and has one main characteristic for better machine learning.
Also, there is the question about local ethical committee protocol for human study (as You made the scans and did not use former database).
Please, write how many scientists made intraoral scan protocol, how many times and what was agreement between them and between tries of each.
Lines 132-133: please, write the variation for ration.
Lines 144-146: please, write the base for number of spheres and their position (previous study, reference etc.)
Lines 208-209: one of the article is Yours as I understood and the conclusion is that rugae could change due to orthodontic treatment. Please, make this fragment clear.
3.3 subsection
Please, explain choice of number of sample for each action for learning. Please, write the base of statement that 80 samples would be enough for adequate machine learning.
Figure 7. Please, add explanation for legend.
Also, please, explain the colors on models/layers for each figure for better understanding.
Please, write the limitations of Your study and method after discussion. Also, please, write if You could advise or not Your method for assessment of orthodontic treatment quality.
Sincerely, Reviewer
